

# Preliminary study on phosphate solubilizing *Bacillus subtilis* strain Q3 and *Paenibacillus* sp. strain Q6 for improving cotton growth under alkaline conditions

Maqshoof Ahmad[1], Iqra Ahmad[1], Thomas H. Hilger[2], Sajid M. Nadeem[3], Muhammad F. Akhtar[1], Moazzam Jamil[1], Azhar Hussain[1] and Zahir A. Zahir[4]

[1] Department of Soil Science, University College of Agriculture and Environmental Sciences, The Islamia University of Bahawalpur, Bahawalpur, Punjab, Pakistan
[2] Institute of Agricultural Sciences in the Tropics (Hans-Ruthenberg Institute), University of Hohenheim, Stuttgart, Germany
[3] Department of Soil Science, University of Agriculture Faisalabad, Sub-campus Burewala-Vehari, Pakistan, Burewala, Punjab, Pakistan
[4] Institute of Soil and Environmental Sciences, University of Agriculture Faisalabad, Faisalabad, Punjab, Pakistan

Corresponding author
Maqshoof Ahmad,
maqshoof_ahmad@yahoo.com

## ABSTRACT

**Background**. Low phosphorus availability limits crop production in alkaline calcareous soils in semi-arid regions including Pakistan. Phosphate solubilizing bacteria may improve crop growth on alkaline calcareous soils due to their ability to enhance P availability.

**Methods**. Twenty rhizobacterial isolates (Q1–Q20) were isolated from rhizosphere of cotton and characterized for their growth promoting attributes *in vitro*. The selected phosphate solubilizing isolates were further screened for their ability to improve cotton growth under axenic conditions (jar trial). The phosphorus solubilization capacities of selected strains were quantified and these strains were identified through 16S rDNA sequencing.

**Results**. Isolates Q2, Q3, Q6, Q7, Q8, Q13 and Q14 were able to solubilize phosphate from insoluble sources. Most of these isolates also possessed other traits including catalase activity and ammonia production. The growth promotion assay showed that Q3 was significantly better than most of the other isolates followed by Q6. Maximum root colonization ($4.34 \times 10^6$ cfu g$^{-1}$) was observed in case of isolate Q6 followed by Q3. The phosphorus solubilization capacities of these strains were quantified, showing a maximum phosphorus solubilization by Q3 (optical density $2.605 \pm 0.06$) followed by the Q6 strain. The strain Q3 was identified as *Bacillus subtilis* (accession # KX788864) and Q6 as *Paenibacillus* sp. (accession # KX788865) through 16S rDNA sequencing.

**Discussion**. The bacterial isolates varied in their abilities for different growth promoting traits. The selected PGPR *Bacillus subtilis* strain Q3 and *Paenibacillus* sp. strain Q6 have multifarious growth promoting traits including ability to grow at higher EC and pH levels, and phosphorus solubilizing ability. These strains can efficiently colonize cotton roots under salt affected soils and help plants in phosphorus nutrition. It is concluded that both strains are potential candidates for promoting cotton growth under alkaline conditions, however further investigation is required to determine their potential for field application.

## INTRODUCTION

Phosphorus limits crop productivity worldwide (*Singh et al., 2013*) including Pakistan. The synchronization of phosphorus supply with plant requirements at critical stages of growth is crucial for optimal crop production (*Lukowiak, Grzebisz & Sassenrath, 2016*). Pakistan soils are alkaline calcareous in nature, thus salinity is often a major problem in cotton growing areas where such soils are vulnerable to P deficiency through fixation and precipitation. Phosphorus deficiency in cotton affects root and shoot growth as well as early boll development. Additionally, P deficiency also prolongs ripening, decreases water use efficiency, and hampers energy storage and transfer within the plant (*Saleem et al., 2010*). In response, farmers usually apply large amounts of fertilizers to improve the growth and yield of cotton. The increasing costs of chemical fertilizers in Pakistan along with negative effects on environment force the researchers to develop some alternate technology that may limit the use of phosphatic fertilizers.

Plant growth promoting rhizobacteria (PGPR) can make P available to crop plants (*Rajput et al., 2013*) and also have other beneficial traits. These bacteria convert insoluble phosphate into soluble form through various processes like production of phosphatases and low molecular weight organic acids, acidification and chelation (*Charest, Beauchamp & Antun, 2005*; *Barroso, Pereira & Nahas, 2006*). Phosphate solubilizing bacteria (PSB) make available the indigenous insoluble phosphate in soil thus reduce the dependence on chemical fertilizers. The PSB can use various other mechanisms for crop improvement such as: (1) nitrogen fixation and mineral nutrient solubilization, (2) suppression of pathogens (by production of hydrogen cyanide, antibiotics, and/or competition for nutrients), (3) improvement of plant tolerances against stresses like drought, salinity, and heavy metal toxicity, and (4) production of phytohormones such as indole-3-acetic acid (IAA) (*Gupta, Gopal & Tilak, 2000*; *Fernando, Nakkeeran & Zhang, 2006*). Some aerobic microorganisms including plant growth promoting rhizobacteria from the genera *Pseudomonas*, *Bacillus* and *Paenibacillus* produce catalase enzyme (*Mumtaz et al., 2017*) that protect them against reactive oxygen species such as hydrogen peroxide. Inoculation of crop plants with PGPR having catalase activity under stressed conditions can be beneficial to neutralize the effect of hydrogen peroxide on crops. Bacteria contain certain putative ammonia-producing enzymes such as ammonia lyases, nitrite reductases, nitrilases, pyridoxamine phosphate oxidases, and amino acid and nucleotide deaminases. These enzymes can produce ammonia from a number of substrates such as nitrite (*Simon, 2002*), amino acids and protein (*Ozugul & Ozugul, 2007*) and urea (*Kleiner, Traglauer & Domm, 1998*) which strongly support the process of ammonia synthesis. Microbially produced ammonia can be beneficial or toxic for plants co-culturing with these microbes (*Weise, Kai & Piechulla, 2013*) that entirely depends upon the initial available soil nitrogen concentrations.

The PSB isolated from alkaline soils can tolerate higher levels of salts (*Ahmad et al., 2011*) thus can induce tolerance in crop plants against adverse environmental stresses

such as drought (*Zahir et al., 2008*), salt (*Ahmad et al., 2011*), weed infestation (*Babalola, 2010*), heavy metal contaminations and nutrient deficiency (*Sheng, 2005*). These bacteria can be more useful to enhance crop production on degraded soils (*Yang, Kloepper & Ryu, 2009*; *Maheshwari et al., 2012*). When selecting the best-suited PGPR for a given site, a combination of two or more traits is a much more promising approach compared to the use of a single character (*Zahir, Arshad & Frankenberger, 2003*; *Ahmad, Ahmad & Khan, 2008*; *Baez-Rogelio et al., 2017*).

To test the hypothesis, current study was conducted to isolate, screen and characterize salt tolerant, P solubilizing PGPR from alkaline calcareous soils for the improvement of cotton productivity.

## MATERIALS AND METHODS

### Isolation of rhizobacteria from cotton rhizosphere

Soil samples were collected from the rhizosphere of cotton plants cultivated on farmer fields at two sites (Bahawalpur and Haroonabad). Isolation of bacterial colonies was performed using the dilution plate technique described by *Dworkin & Foster (1958)*. For purification of bacterial colonies, samples were streaked onto general-purpose agar plates prepared from glucose (10 g), $K_2HPO_4$ (2.5 g), $KH_2PO_4$ (2.5 g), $(NH_4)_2HPO_4$ (1.0 g), $MgSO_4.7H_2O$ (0.2 g), $FeSO_4.7H_2O$ (0.01 g), $MnSO_4.7H_2O$ (0.007 g) and agar powder (15 g) per liter of deionized water and final pH adjusted to 7.2. These plates were incubated at 30 °C for 2 days. Isolated single colonies were removed and re-streaked to fresh general purpose agar plate and incubated as described above. The technique was repeated and the single colony cultures were preserved for further experimentation.

### Characterization

The bacterial isolates were characterized for catalase activity as described by *MacFaddin (1980)*. A urease test was carried out using sterilized urea broth (*Shruti, Arun & Yuvneet, 2013*). The ability of rhizobacterial strains to solubilize inorganic phosphate was investigated using Pikovskaya's agar medium (*Pikovskaya, 1948*). The ammonia production test was performed using Nessler's reagent, following methods by *Cappuccino & Sherman (1992)*. The Gram staining and cell shape of bacteria was studied after 48 h of growth on agar plates (*Vincent, 1970*).

### Osmoadaptation assay

Osmoadaptation assays of bacterial isolates were performed as described by *Zahir et al. (2010)*. Salinity tolerance of bacterial isolates was assessed at 1.42 (control), 4, 8 and 12 dS $m^{-1}$ salinity levels developed by adding calculated amounts of mixed salts (NaCl, $MgSO_4$, $CaCl_2$ and $Na_2SO_4$). Salts were calculated by using quadratic equation as described by *Haider & Ghafoor (1992)* for the development of salinity levels in broth. The broth was autoclaved at 121 °C for 30 min and then 25 mL of broth was taken in 50 mL flasks and inoculated with rhizobacterial isolates. Flasks were incubated at 30 °C and after 3 days of incubation, absorbance (optical density) was measured by using spectrophotometer (Model Carry 60; Agilent Tech., Santa Clara, CA, USA) at 600 nm wavelength.

## Growth of PGPR at different pH levels

The growth of bacterial isolates was determined at three pH levels using methods by *Shruti, Arun & Yuvneet (2013)*. The purpose of the study was to assess the ability of isolates to grow under alkaline conditions. Nutrient broth was poured into test tubes and pH was adjusted to 4.0, 7.0 and 10.0. The tubes were autoclaved, cooled and inoculated with the respective strain and incubated at 30 °C for 24 h. The optical density (OD) was measured by using a spectrophotometer (Model Carry 60; Agilent Tech., Santa Clara, CA, USA) at 600 nm wavelength.

## Root colonization assay

Root colonization ability of phosphate solubilizing isolates in cotton was studied under axenic conditions, following methods by *Simons et al. (1996)*. Glass jars were sterilized and filled with sterilized sand. Half strength Hoagland solution was used to moisten the sand. Surface-sterilized cotton seeds were submerged for ten minutes in the broth of the respective strains and were then sown. Jars were placed in growth chambers at 28 $\pm$ 1 °C. After 1 week, root tips (0.2 g) were removed and added to 5 mL sterilized distilled water and shaken vigorously for 30 min using an orbital shaker (Memmert, Schwabach, Germany) at 100 rpm. Bacterial suspension solutions were made from $10^{-1}$ to $10^{-6}$. One mL of each dilution was poured onto petri plates filled with sterilized general purpose media that were then incubated at 28 $\pm$ 1 °C. The CFU mL$^{-1}$ was calculated and colonies were counted by using a digital colony counter (J.P Selecta, Barcelona, Spain).

## Quantitative estimation of phosphate solubilization

Cultures showing positive results in the agar medium examinations were further assessed for P solubilization quantitatively using methods described by *Clescerie, Greenberg & Eaton (1998)*. For this purpose, cultures were inoculated in 50 mL of Pikovskaya's broth and incubated at 30 °C for 48 h. The broth was centrifuged and 5 mL of the supernatant was taken following by addition of 5 mL of Vanadomolybdate solution. The volume was made up to 25 mL and incubated overnight for development of yellow color. The absorbance was measured by using spectrophotometer (Model Carry 60, Agilent Tech., Santa Clara, CA, USA) at 420 nm wavelength.

## Evaluation of phosphate solubilizing PGPR under axenic conditions

Based on *in vitro* characterization of PGPR strains, seven phosphate solubilizing strains were selected for their plant growth promoting abilities under axenic conditions. The experiment was conducted at College of Agriculture and Environmental Sciences, the Islamia University of Bahawalpur. Cotton seeds were surface sterilized by submersion in 95% ethanol for 30 s, followed by submersion in 0.2% HgCl$_2$ solution for 3 min. Seeds were then washed thoroughly with sterilized water. Three surface-sterilized seeds were inoculated in each of the seven selected isolates by submerging them in the respective broth for ten minutes. In case of the control, surface-sterilized seeds were treated with sterilized broth without inoculation. Inoculated seeds were transplanted to autoclaved glass jars filled with sand. Sterilized Hoagland solution (*Hoagland & Arnon, 1950*) (modified by adding 1.5 g L$^{-1}$ phosphate rock (Sigma-Aldrich, St. Louis, MO, USA) as source of

**Table 1  Characterization of rhizobacterial isolates.**

| Isolate | Gram staining | Cell shape | Catalase reaction | Urease production | Phosphate solubilization | Ammonia production |
|---------|--------------|-----------|-------------------|-------------------|--------------------------|--------------------|
| Q1 | +ve | Rod | + | − | − | − |
| Q2 | −ve | Rod | + | − | + | + |
| Q3 | +ve | Rod | + | − | + | + |
| Q4 | +ve | Rod | + | − | − | − |
| Q5 | −ve | Rod | + | − | − | − |
| Q6 | +ve | Rod | + | − | + | + |
| Q7 | +ve | Rod | + | − | + | − |
| Q8 | −ve | Rod | + | − | + | + |
| Q9 | +ve | Rod | + | − | − | − |
| Q10 | −ve | Rod | + | − | − | − |
| Q11 | −ve | Rod | + | − | − | − |
| Q12 | +ve | Rod | + | − | − | − |
| Q13 | −ve | Rod | + | − | + | + |
| Q14 | +ve | Rod | + | − | + | + |
| Q15 | −ve | Rod | + | − | − | + |
| Q16 | −ve | Rod | + | − | − | − |
| Q17 | +ve | Rod | + | − | − | − |
| Q18 | −ve | Rod | + | − | − | + |
| Q19 | +ve | Rod | + | + | − | + |
| Q20 | +ve | Rod | + | − | − | − |

**Notes.**

Growth = (+), No growth = (−).

phosphorus instead of $KH_2PO_4$) was added in the jars to provide nutrients to the seedlings. Jars were arranged in a completely randomized design (CRD) with three replications in a growth room with controlled temperature (30 °C), humidity (55–67%) and light intensity (1,300–1,400 μmoles/m$^2$/s) with a 16 h light and 8 h dark cycle. Growth parameters, i.e., shoot length, root length, shoot fresh weight, root fresh weight, shoot dry weight, root dry weight and root/shoot ratio, were recorded after 20 days of germination.

## 16S rDNA sequencing of selected strains

Two strains (Q3 and Q6) were selected and identified through amplification, sequencing and bioinformatics analysis of their 16S rDNA gene sequences. For this purpose, crude DNA of the selected isolates Q3 and Q6 was extracted from the cell culture using proteinase K treatment (*Cheneby et al., 2004*). The 16S rDNA sequence was amplified in a thermocycler (Eppendorf, Hauppauge, NY, USA) using the universal primers for forward 785F (5′-GGATTAGATACCCTGGTA-3′) and reverse 907R (5′-CCGTCAATTCMTTTRAGTTT-3′) reactions. The PCR reaction was carried out using 2.5 μL crude DNA as a template following the program as described by *Hussain et al. (2011)*. The size of the amplified 16S rDNA was confirmed by separation on 1% agarose gel along with GeneRuler 1kb DNA (Fermentas, Burlington, Canada). The 16S rDNA PCR product was purified using a PCR Purification Kit (Favorgen, Changzhi Township, Taiwan) and amplified PCR

**Table 2  Growth (optical density) of rhizobacterial isolates at different salinity levels.**

| Isolate | Salinity level (dS m$^{-1}$) | | | |
|---|---|---|---|---|
| | 1.42 (control) | 4.0 | 8.0 | 12.0 |
| Q1 | 0.764 ± 0.0015 | 0.663 ± 0.0021 | 0.526 ± 0.0036 | 0.512 ± 0.0046 |
| Q2 | 0.834 ± 0.0055 | 0.723 ± 0.0052 | 0.614 ± 0.0015 | 0.513 ± 0.0031 |
| Q3 | 0.732 ± 0.0028 | 0.624 ± 0.0020 | 0.656 ± 0.0012 | 0.696 ± 0.0011 |
| Q4 | 0.638 ± 0.0030 | 0.686 ± 0.0052 | 0.764 ± 0.0031 | 0.523 ± 0.0058 |
| Q5 | 0.528 ± 0.0068 | 0.521 ± 0.0025 | 0.493 ± 0.0030 | 0.433 ± 0.0055 |
| Q6 | 0.844 ± 0.0072 | 0.792 ± 0.0081 | 0.815 ± 0.0011 | 0.751 ± 0.0037 |
| Q7 | 0.834 ± 0.0015 | 0.833 ± 0.0030 | 0.771 ± 0.0076 | 0.622 ± 0.0036 |
| Q8 | 0.713 ± 0.0081 | 0.604 ± 0.0017 | 0.623 ± 0.0030 | 0.542 ± 0.013 |
| Q9 | 0.643 ± 0.0097 | 0.593 ± 0.0026 | 0.511 ± 0.0023 | 0.514 ± 0.0025 |
| Q10 | 0.593 ± 0.0011 | 0.554 ± 0.0062 | 0.523 ± 0.0015 | 0.512 ± 0.0035 |
| Q11 | 0.664 ± 0.0060 | 0.631 ± 0.0018 | 0.543 ± 0.002 | 0.634 ± 0.0025 |
| Q12 | 0.783 ± 0.0062 | 0.509 ± 0.0043 | 0.571 ± 0.0063 | 0.652 ± 0.0024 |
| Q13 | 0.922 ± 0.011 | 0.845 ± 0.0015 | 0.713 ± 0.0012 | 0.453 ± 0.0028 |
| Q14 | 0.643 ± 0.0011 | 0.623 ± 0.0055 | 0.643 ± 0.0015 | 0.602 ± 0.0088 |
| Q15 | 0.714 ± 0.0099 | 0.723 ± 0.013 | 0.506 ± 0.0076 | 0.524 ± 0.0087 |
| Q16 | 0.574 ± 0.0172 | 0.553 ± 0.013 | 0.484 ± 0.0073 | 0.466 ± 0.0037 |
| Q17 | 0.603 ± 0.0045 | 0.544 ± 0.0051 | 0.453 ± 0.003 | 0.432 ± 0.010 |
| Q18 | 0.693 ± 0.0023 | 0.624 ± 0.0017 | 0.533 ± 0.0072 | 0.392 ± 0.0028 |
| Q19 | 0.734 ± 0.004 | 0.642 ± 0.0028 | 0.513 ± 0.0015 | 0.536 ± 0.0045 |
| Q20 | 0.615 ± 0.0047 | 0.542 ± 0.0028 | 0.643 ± 0.007 | 0.523 ± 0.0018 |

**Notes.**
Mean ± standard error.

products were sequenced using a commercial service offered by Macrogen Seoul, Korea (http://macrogen.com/eng/).

Strains were identified using a partial sequence of the 16S rDNA gene on MEGA 7.0.14 software (Pennsylvania State University, USA) and BLASTn search on NCBI servers. Sequences of closely related and validly published type strains ($n = 15$) used for constructing the phylogenetic tree were selected and retrieved from the MEGA database. The phylogenetic and molecular analyses were performed with selected closely related taxa according to procedure using MEGA version 7.0.14 (*Kumar, Stecher & Tamura, 2016*). The evolutionary history was inferred using the Neighbor-Joining method (*Saitou & Nei, 1987*). The percentage of replicate trees in which the associated taxa clustered together in the bootstrap test (500 replicates) is shown next to the branches (*Felsenstein, 1985*). The evolutionary distances were computed using the Maximum Composite Likelihood method (*Tamura, Nei & Kumar, 2004*) and are in the units of the number of base substitutions per site. All positions containing gaps and missing data were eliminated. The stability of the relationship was assessed by bootstrap analysis by performing 500 re-samplings for the tree topology of the neighbour-joining method L 195. We replaced "but" with "with".

**Table 3** Growth (optical density) of rhizobacterial isolates at different pH levels.

| Isolate | pH level | | |
|---|---|---|---|
| | **4.0** | **7.0** | **10.0** |
| Q1 | $0.193 \pm 0.0037$ | $0.764 \pm 0.0015$ | $0.822 \pm 0.005$ |
| Q2 | $0.252 \pm 0.004$ | $0.834 \pm 0.0055$ | $0.743 \pm 0.007$ |
| Q3 | $0.221 \pm 0.0034$ | $0.732 \pm 0.0028$ | $0.712 \pm 0.006$ |
| Q4 | $0.184 \pm 0.002$ | $0.638 \pm 0.003$ | $0.794 \pm 0.006$ |
| Q5 | $0.152 \pm 0.0062$ | $0.528 \pm 0.007$ | $0.633 \pm 0.0066$ |
| Q6 | $0.181 \pm 0.0061$ | $0.844 \pm 0.0072$ | $0.743 \pm 0.0057$ |
| Q7 | $0.194 \pm 0.0028$ | $0.834 \pm 0.0015$ | $0.726 \pm 0.009$ |
| Q8 | $0.129 \pm 0.0057$ | $0.713 \pm 0.0081$ | $0.614 \pm 0.0047$ |
| Q9 | $0.236 \pm 0.0035$ | $0.643 \pm 0.0097$ | $0.693 \pm 0.002$ |
| Q10 | $0.143 \pm 0.003$ | $0.593 \pm 0.0011$ | $0.732 \pm 0.0057$ |
| Q11 | $0.106 \pm 0.0015$ | $0.664 \pm 0.006$ | $0.793 \pm 0.0026$ |
| Q12 | $0.181 \pm 0.0054$ | $0.783 \pm 0.0062$ | $0.737 \pm 0.0005$ |
| Q13 | $0.164 \pm 0.0017$ | $0.922 \pm 0.0118$ | $0.914 \pm 0.004$ |
| Q14 | $0.183 \pm 0.0011$ | $0.643 \pm 0.0011$ | $0.853 \pm 0.0023$ |
| Q15 | $0.137 \pm 0.0014$ | $0.714 \pm 0.0099$ | $0.624 \pm 0.0035$ |
| Q16 | $0.246 \pm 0.001$ | $0.574 \pm 0.0172$ | $0.795 \pm 0.0014$ |
| Q17 | $0.113 \pm 0.0011$ | $0.603 \pm 0.0045$ | $0.714 \pm 0.0055$ |
| Q18 | $0.152 \pm 0.003$ | $0.693 \pm 0.0023$ | $0.682 \pm 0.0055$ |
| Q19 | $0.174 \pm 0.0026$ | $0.734 \pm 0.004$ | $0.747 \pm 0.0043$ |
| Q20 | $0.163 \pm 0.006$ | $0.615 \pm 0.0047$ | $0.721 \pm 0.0091$ |

**Notes.**
Mean $\pm$ standard error.

**Table 4** Root colonization ability of phosphate solubilizing PGPR.

| Rhizobacterial isolate | Q2 | Q3 | Q6 | Q7 | Q8 | Q13 | Q14 |
|---|---|---|---|---|---|---|---|
| Root colonization (cfu g$^{-1}$) | $4.62 \times 10^5$ | $4.34 \times 10^6$ | $3.76 \times 10^6$ | $3.62 \times 10^6$ | $4.46 \times 10^5$ | $3.86 \times 10^4$ | $4.74 \times 10^5$ |

## Statistical analysis

The data obtained from above tests were subjected to statistical analysis using MS Excel (Microsoft Office 10) and analysis of variance techniques (ANOVA) in Statistix 8.1. The means were compared using LSD at 5% probability level ($p \leq 0.05$) (*Steel, Torrie & Dicky, 1997*).

# RESULTS

## Characterization

All 20 strains we isolated were positive for catalase. One strain (Q19) was positive for urease. Seven strains, i.e., Q2, Q3, Q6, Q7, Q8, Q13 and Q14, formed a clear zone around the colony in Pikovskaya agar medium and were found to show phosphorus solubilization capacities. Nine strains, i.e., Q2, Q3, Q6, Q8, Q13, Q14, Q15, Q18 and Q19, possessed the ability to produce ammonia. Eleven strains including Q3 and Q6 showed positive results for Gram staining while all the tested strains were rod shaped bacteria (Table 1).

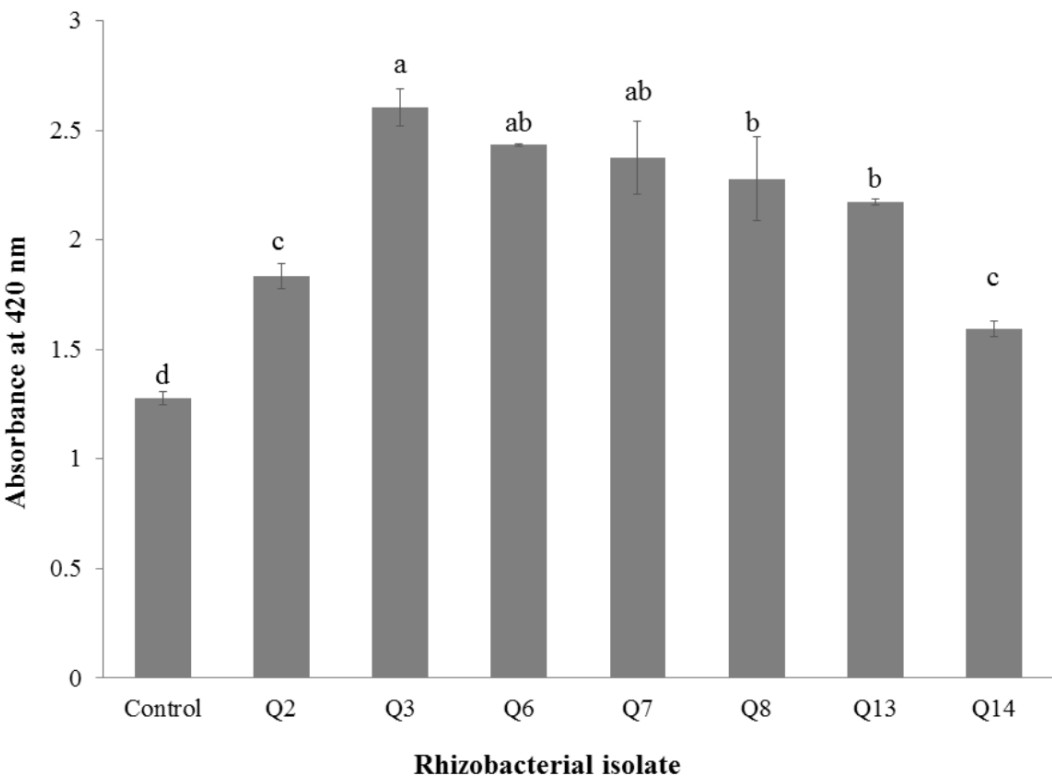

**Figure 1** **Phosphate solubilization by selected strains under liquid broth ($n = 3$).** Bars with the same letters are statistically non-significant at the 5% probability level.

Salinity tolerance of rhizobacterial strains varied among tested strains (Table 2). As the salinity level increased, most of the rhizobacterial growth decreased. At the highest NaCl salinity level (12 dS m$^{-1}$), the maximum optical density was given by strain Q6 followed by Q3, Q12 and Q11. The data reveals that maximum growth (OD at 620 nm) of bacterial isolates occurred at neutral and alkaline pH (Table 3). All rhizobacterial isolates showed poor growth under acidic pH. At neutral pH, the bacterial isolate Q13 showed maximum growth followed by Q6, Q7, Q2 and Q12. At alkaline pH (10.0), Q13 showed maximum growth followed by Q14 and Q1.

All selected strains efficiently colonized cotton roots with the maximum root colonization ($4.34 \times 10^6$ cfu g$^{-1}$) was observed in the case of strain Q3 followed by Q6 (Table 4). All strains were solubilizing inorganic phosphate in liquid broth (Fig. 1) but maximum optical density was given by Q3 ($2.605 \pm 0.06$) followed by Q6 ($2.44 \pm 0.003$) and Q7 ($2.38 \pm 0.12$).

### Effect of phosphate solubilizing PGPR on the growth of cotton seedlings

The seven isolates with phosphate solubilizing capacities that showed potential growth at the higher salinity levels were selected to investigate their plant growth promoting activities

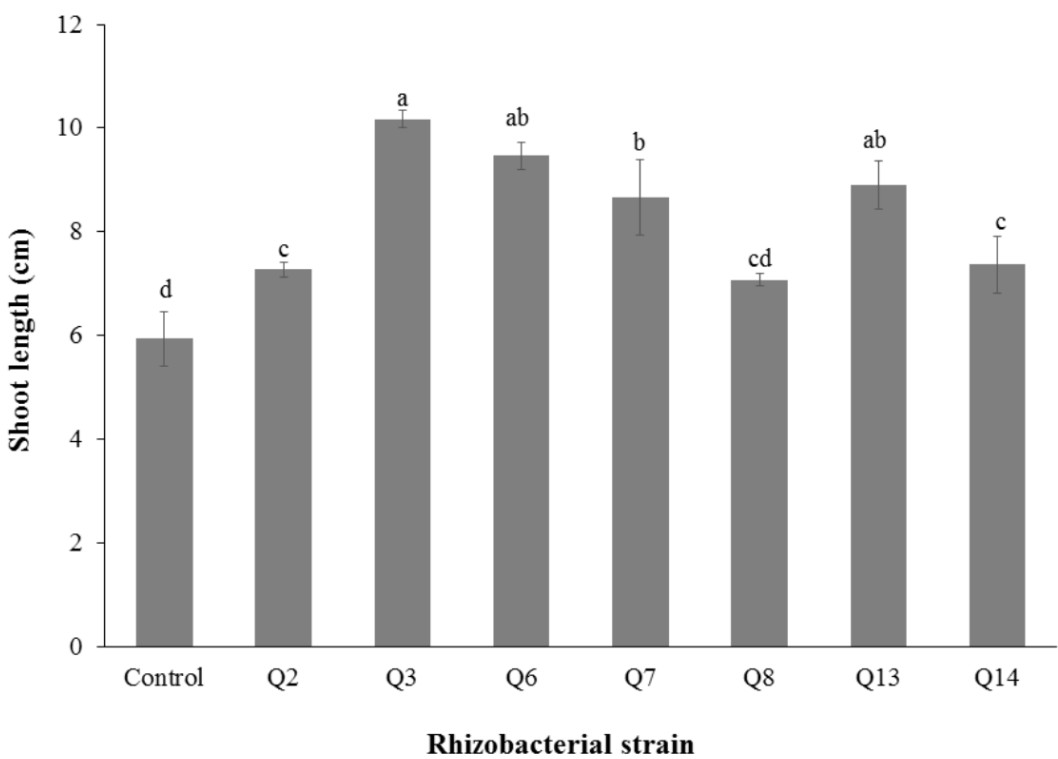

**Figure 2** **Effect of phosphate solubilizing rhizobacteria on the shoot length of cotton seedlings.** Bars with the same letters are statistically non-significant at the 5% probability level.

for cotton seedlings under axenic conditions. Results indicated that inoculation with these strains increased the growth of cotton seedlings as compared to the un-inoculated control.

Inoculation of phosphate solubilizing rhizobacteria significantly enhanced the shoot length (Fig. 2). Increase in shoot length was 71%, 59%, 50%, 46%, 24%, and 22% for Q3, Q6, Q13, Q7, Q14 and Q2, respectively, as compared to the un-inoculated control. Increases in the root length of the cotton seedlings were promising for all strains (Fig. 3). The maximum increase in the root length was observed with strain Q3 (162%), followed by Q6, Q13, Q8 and Q7. Phosphate solubilizing rhizobacteria significantly increased the shoot fresh weight of cotton seedlings compared to the control (Fig. 4). The maximum increase in the shoot fresh weight was observed with strain Q6 (87%), followed by Q3 (75%), Q7 (63%) and Q13 (60%) as compared to the un-inoculated control. The results of three isolates, i.e., Q2, Q8 and Q14, were non-significant when compared with control.

Inoculation of phosphate solubilizing rhizobacteria promoted a significant enhancement in the root fresh weight as compared to the un-inoculated control (Fig. 5). Maximum increase in the root fresh weight was observed with strain Q3 (3 fold) followed by Q6 (277%), Q7 (176%) and Q2 (150%), which were significantly higher than un-inoculated control. The two strains Q8 and Q14 were found to be non-significant compared with the control. The results regarding the root/shoot ratio revealed that phosphate solubilizing rhizobacterial strains contributed to enhancement of the root shoot ratio. The strain Q3

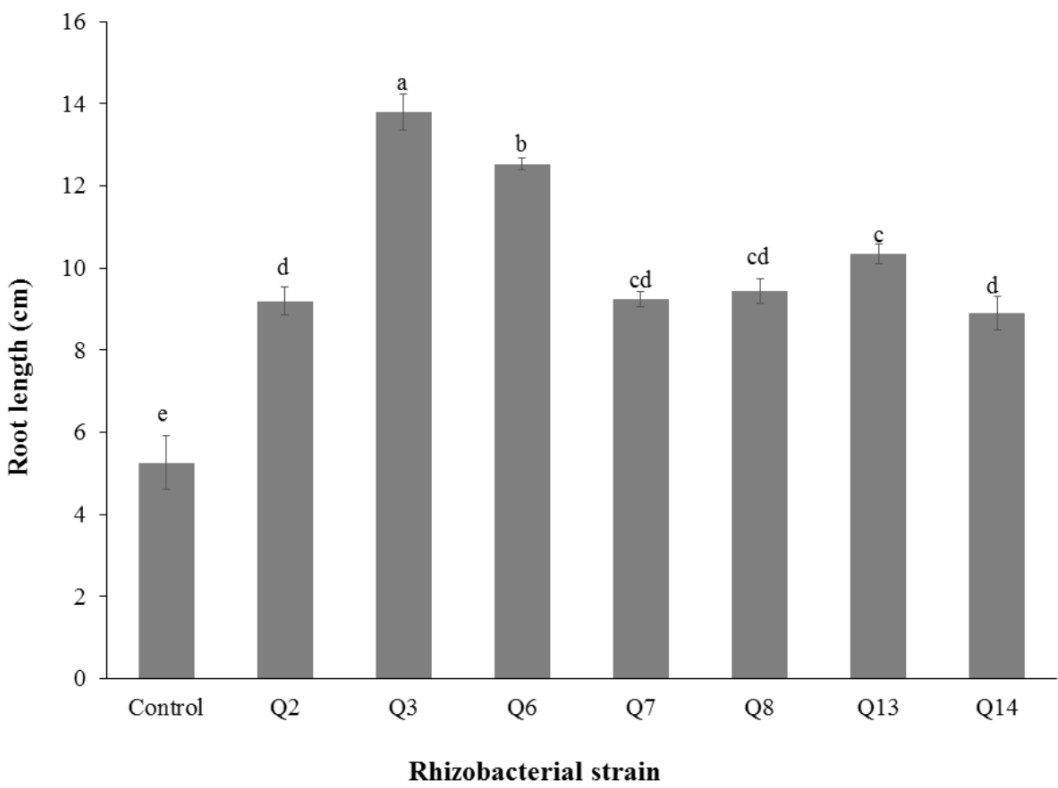

**Figure 3  Effect of phosphate solubilizing rhizobacteria on the root length of cotton seedlings.** Bars with the same letters are statistically non-significant at the 5% probability level.

showed maximum increase in root/shoot ratio (50%) followed by Q6 and Q8 compared to the un-inoculated control (Fig. 6).

Shoot dry weight was significantly improved by rhizobacterial strains except Q2 where the increase was non-significant when compared with control. Maximum shoot dry weight of cotton seedlings was improved by Q6 followed by Q3 (Fig. 7). The PSB strains also improved the root dry weight of cotton seedlings however this increase was non-significant in case of Q2, Q8, Q13 and Q14 when compared with control. Maximum root dry weight was observed due to inoculation with Q3 followed by Q6 and Q7 where the results were significantly better than in-inoculated control (Fig. 8).

## Identification of selected strains through 16S rDNA sequencing

The 16S rDNA genes 1000 bp and 1203 bp amplified from the strains Q3 and Q6 were sequenced and the sequence results were deposited in the GenBank database under the accession numbers KX788864 and KX788865 for Q3 and Q6, respectively.

The BlastN analysis of the 16S rDNA amplicon indicated their maximum similarity with the bacterial strains belonging to genus *Bacillus* and *Paenibacillus,* respectively. The analysis of the 16S rDNA of the bacterial strains Q3 and Q6 was carried out by constructing the phylogenetic tree following the neighbor joining method (Figs. 9 and 10). The bacterial strains Q3 and Q6 were observed to be phylogenetically positioned in the

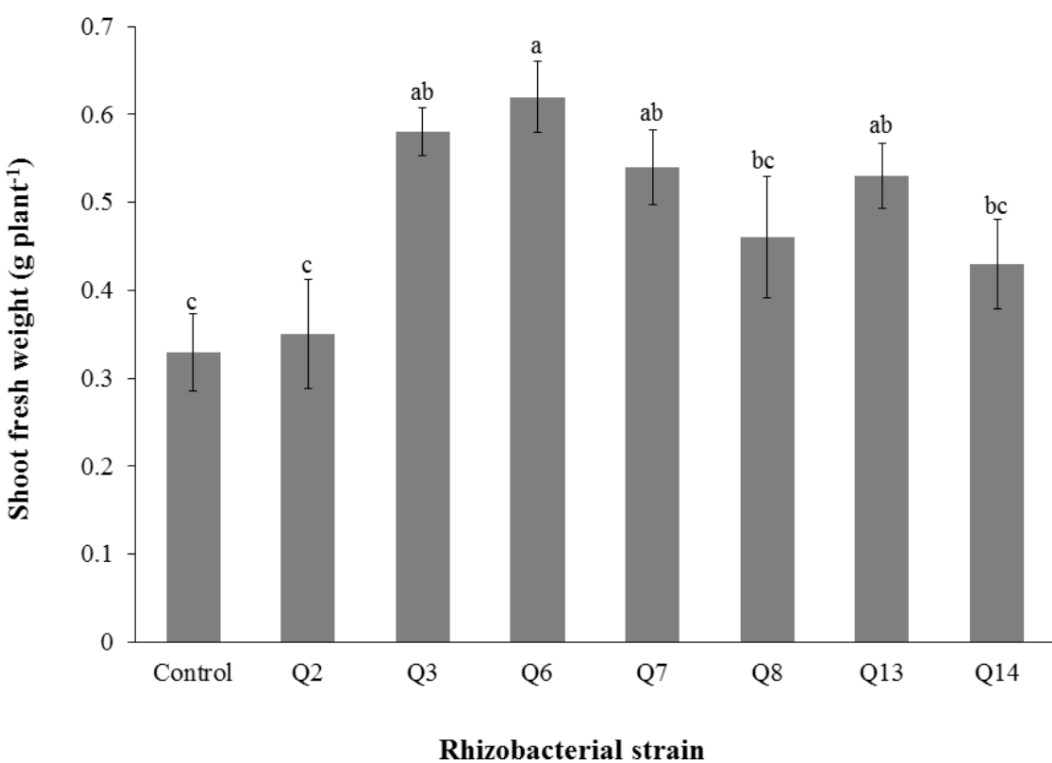

**Figure 4** **Effect of phosphate solubilizing rhizobacteria on the shoot fresh weight of cotton seedlings.** Bars with the same letters are statistically non-significant at the 5% probability level.

cluster comprising the bacterial strains belonging to the genus *Bacillus* and *Paenibacillus*, respectively. Following the phylogenetic relationship of the strain Q3 with *B. subtilis* (Fig. 9) and Q6 with several *Paenibacillus* sp. (Fig. 10), these bacterial isolates were named *Bacillus subtilis* strain Q3 and *Paenibacillus* sp. strain Q6, respectively.

## DISCUSSION

This work describes the isolation, characterization and identification of plant growth promoting rhizobacteria from the rhizosphere of cotton for plant growth promoting traits. The PGPR produce catalase enzyme (*Mumtaz et al., 2017*) that helps plants to withstand under stressed conditions by neutralizing the effect of hydrogen peroxide. In the present study, all tested isolates were catalase positive, which is correlated with previous work as described by *Chaiharn et al. (2008)* and *Bumunang & Babalola (2014)*. Similarly, *Shruti, Arun & Yuvneet (2013)* isolated *Fluorescent Pseudomonas* strains from *Oryza sativa* and found that all were catalase positive. In another study, positive results for catalase tests by all isolates have also been reported by *Shrivastava (2013)*.

Strain Q19 was positive for the urease test. Biochemical tests reported by *Shrivastava (2013)* also showed positive results for the urease test. *Rana et al. (2011)* observed that out of ten strains, only five were urease positive. In this study, nine strains were observed to produce ammonia, indicating their ability to provide ammonia to plants as a nutrient

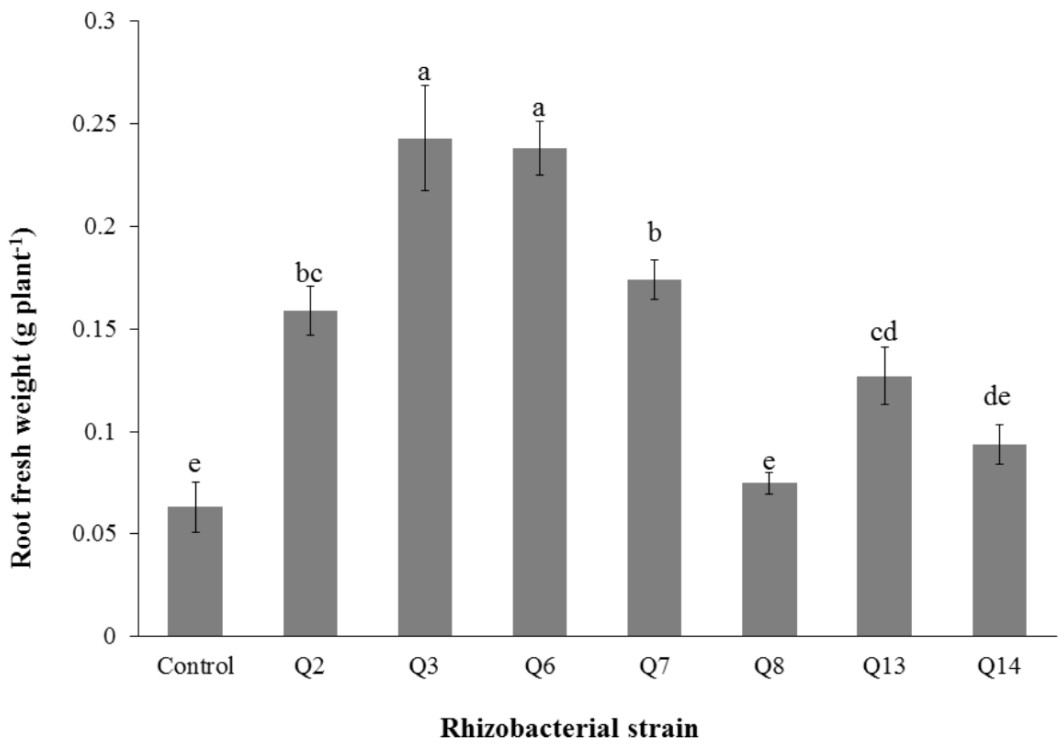

**Figure 5 Effect of phosphate solubilizing rhizobacteria on the root fresh weight of cotton seedlings.**
Bars with the same letters are statistically non-significant at the 5% probability level.

source. Ammonia production by the *K. turfanensis* strain 2M4 along with other PGPR strains such as *Bacillus*, *Pseudomonas*, *Rhizobium*, *Azotobacter*, and *Enterobacter* was reported by *Joseph, Patra & Lawerence (2007)* and *Goswami et al. (2014)*.

Phosphate solubilization by rhizobacterial isolates was also observed, indicated by a clear zone around the bacterial colony in Pikovaskaya agar medium. Out of 20 isolates, seven isolates gave positive results for phosphate solubilization. These findings are in accordance with the results of *Bumunang & Babalola (2014)*. Similarly, *Hussain et al. (2013)* found 32 bacterial isolates with phosphorus solubilizing activity. It has been reported that phosphate solubilizing bacteria release organic acids into the soil which increase the acidity of soil environment and result in decrease of pH. Due to the increased acidity, complex forms of phosphorus can be converted into plant usable forms (*Chen et al., 2006*).

The ability of bacteria to grow at higher salinity and pH levels is considered as an indicator of successful use of these isolates under alkaline calcareous soils in arid and semi-arid regions. As the aridity favors the development of high EC/salinity soils so the ability of microbes to grow at high EC and pH levels may help to cope with such conditions. The results of current study indicate that the growth of bacterial isolates varies with salinity level. Most of the isolates showed decreased growth with increasing salinity. This may be due to the adverse or negative impact of NaCl on the PGPR community as demonstrated by *Nelson & Mele (2007)*. According to the authors, NaCl influences the quantity and/or quality of root exudates, and affects the rhizosphere microbial community structure. The

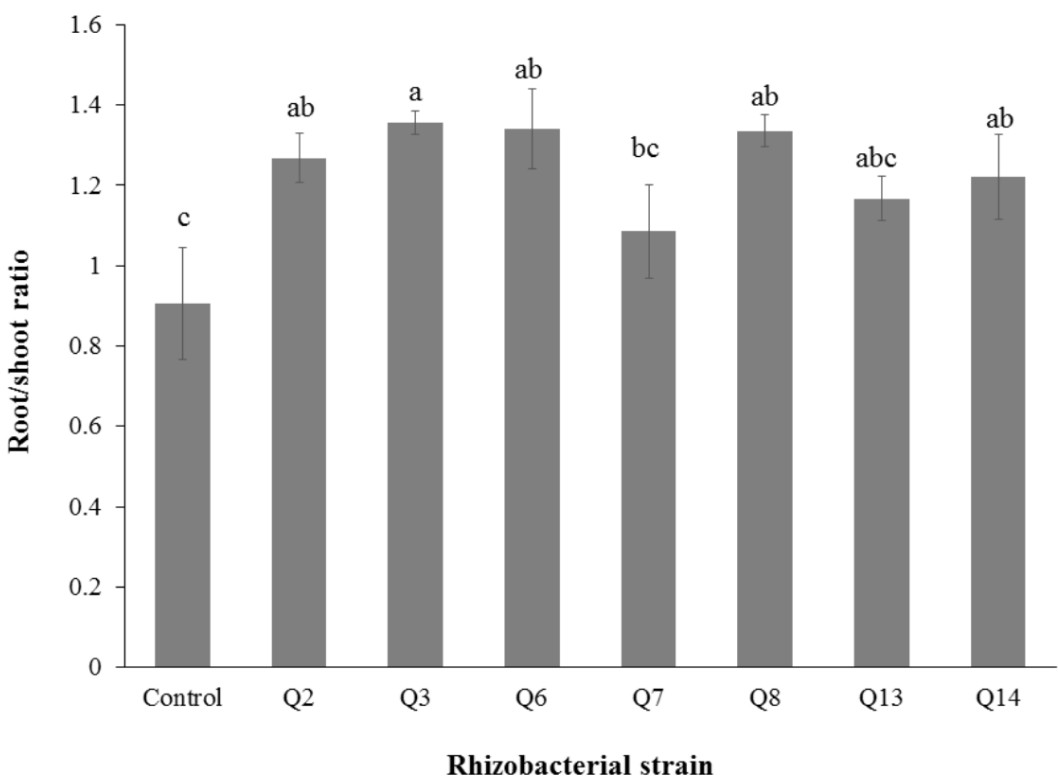

**Figure 6** **Effect of phosphate solubilizing rhizobacteria on root/shoot ratio of cotton seedlings.** Bars with the same letters are statistically non-significant at the 5% probability level of probability.

growth of rhizobacterial isolates at higher salinity level might be due to their tolerance against salinity. The ability of bacterial isolates to grow under salinity stress has been described by *Nakbanpote et al. (2014)*. They reported that bacterial isolates were able to grow in 8% NaCl and *Chakraborty et al. (2014)* found that PGPR can tolerate up to 10% (w/v) salt concentration. The findings from the present study reveal that all of the rhizobacterial isolates showed maximum growth (OD at 620 nm) at neutral and alkaline pH and were negatively affected at an acidic pH. It is well documented that bacteria grow well at neutral to alkaline pH. For example, *Shruti, Arun & Yuvneet (2013)* and *Chakraborty et al. (2014)* reported that bacterial strains showed maximum growth at alkaline pH. According to the authors, bacterial isolates grow within a pH range of 6–12 with maximum growth at pH 7.0. Another study reveals that all isolates were able to grow in a pH range of 6.0–8.5 and 83% of them even tolerated a pH of 10 (*Woyessa & Assefa, 2011*). However, optical density of bacterial isolates significantly decreased at acidic pH levels.

The results show that all of the selected strains were able to colonize the cotton roots and this strengthens their suitability as inoculants. *Ehsan, Saleem & Zafar (2014)* also reported that rhizobacteria have this ability and found a bacterial population of up to $8 \times 10^7$ cfu/g root. *Krzyzanowska et al. (2012)* also observed root colonization potential of various strains on potato roots, however, there are various factors influencing this ability. *Vessey (2003)*, for example, reported that the population density of bacteria in the rhizosphere and roots

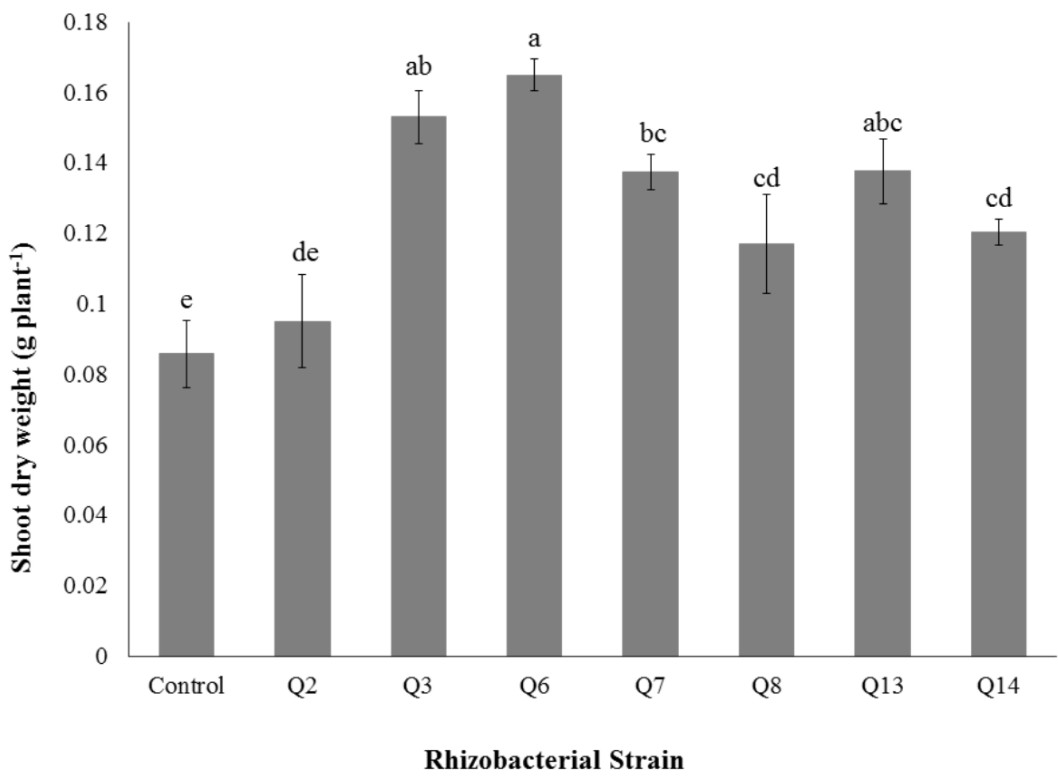

**Figure 7** **Effect of phosphate solubilizing rhizobacteria on the shoot dry weight of cotton seedlings.** Bars with the same letters are statistically non-significant at the 5% probability level.

depends on the nature of the root exudates, root morphology, stages of plant growth, as well as physical and chemical properties of the soil.

In the present study, the salinity tolerant, phosphate solubilizing rhizobacteria significantly increased the growth parameters of cotton seedlings. The results indicate that the rhizobacteria were involved in solubilization of insoluble phosphorus which may have led to the release of organic acids and by the activity of phosphatase enzymes that lowered the pH of medium, thus utilized P as a sole source of phosphorus (*Puente et al., 2004*). Similar results were obtained by *Trivedi et al. (2003)* when PSB (*Bacillus subtilis*) significantly increased the root length of rice in comparison with the control. Similarly, other scientists also observed increased shoot weight by phosphate solubilizing bacteria (*Ehsan, Saleem & Zafar, 2014*). Similar to these results, *Hussain et al. (2013)* found that all bacterial isolates significantly improved the growth of maize plants as compared to an un-inoculated control. In another study, *Ehsan, Saleem & Zafar (2014)* reported that inoculation with phosphate solubilizing PGPR could increase P availability to plants by lowering the pH through the production of organic acids. The authors concluded that the use of phosphate solubilizing bacteria with subsequent organic acid production and phosphatases activity can be a useful tool to improve sustainability in agriculture.

In the present study, the selected strains Q3 and Q6 were identified as *Bacillus subtilis* strain Q3 and *Paenibacillus* sp. strain Q6. These strains are Gram positive, catalase positive,

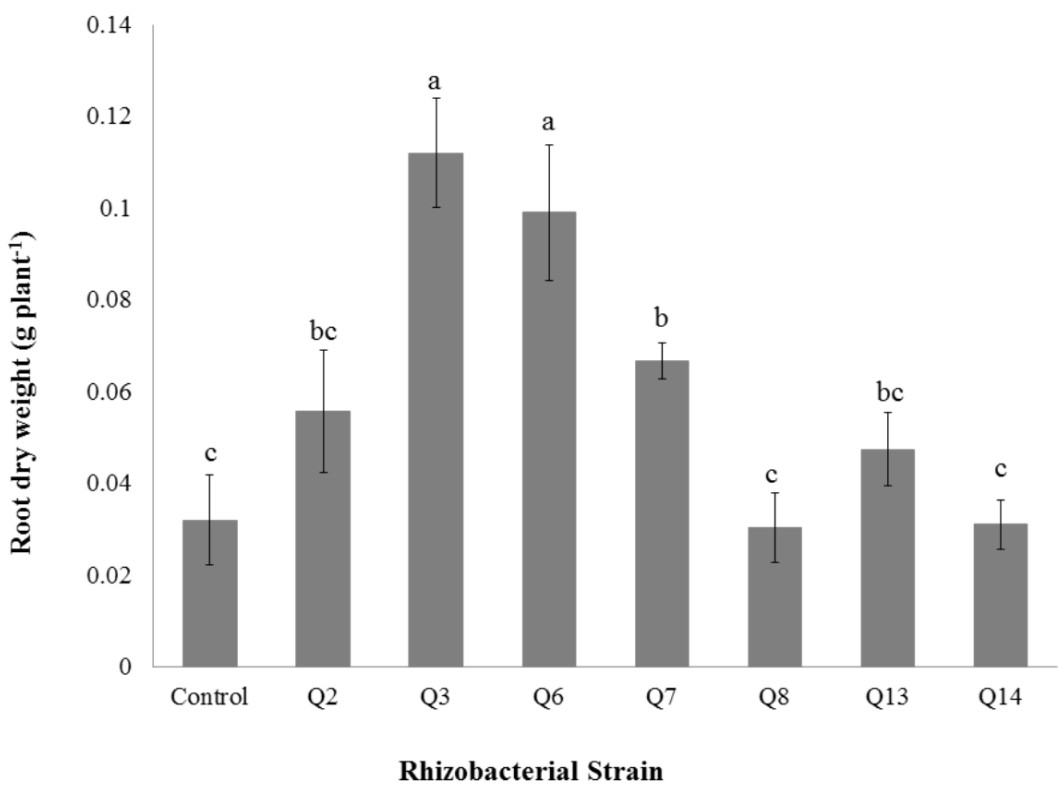

**Figure 8** **Effect of phosphate solubilizing rhizobacteria on the root dry weight of cotton seedlings.** Bars with the same letters are statistically non-significant at the 5% probability level.

salt tolerant and phosphate solubilizing rod shaped bacteria which have the ability to improve root and shoot growth of cotton under axenic conditions. In previous studies, the *Bacillus subtilis* and *Paenibacillus* spp. have been well documented as Gram positive, rod shaped, plant growth promoting rhizobacteria with catalase activity, phosphate solubilization activity, as well as other traits (*Govindasamy et al., 2010*; *Goswami et al., 2015*). Nitrogen fixation abilities of *Paenibacillus* have even been reported (*Xie et al., 2014*). The salinity tolerance and phosphorus solubilization traits of these bacterial strains enhance their scope to be used as inoculants for improving cotton productivity under alkaline calcareous soil conditions such as those prevailing in cotton growing areas of Pakistan.

## CONCLUSIONS

It is concluded that the bacterial isolates varied in their abilities for different growth promoting traits. The selected PGPR *Bacillus subtilis* strain Q3 and *Paenibacillus* sp. strain Q6 have multifarious growth promoting traits including the ability to grow at higher EC and pH levels, and phosphorus solubilizing ability. According to present knowledge, no such strains have yet to be reported that have salinity tolerance and phosphorus solubilization abilities along with root colonization and growth promoting potential especially for

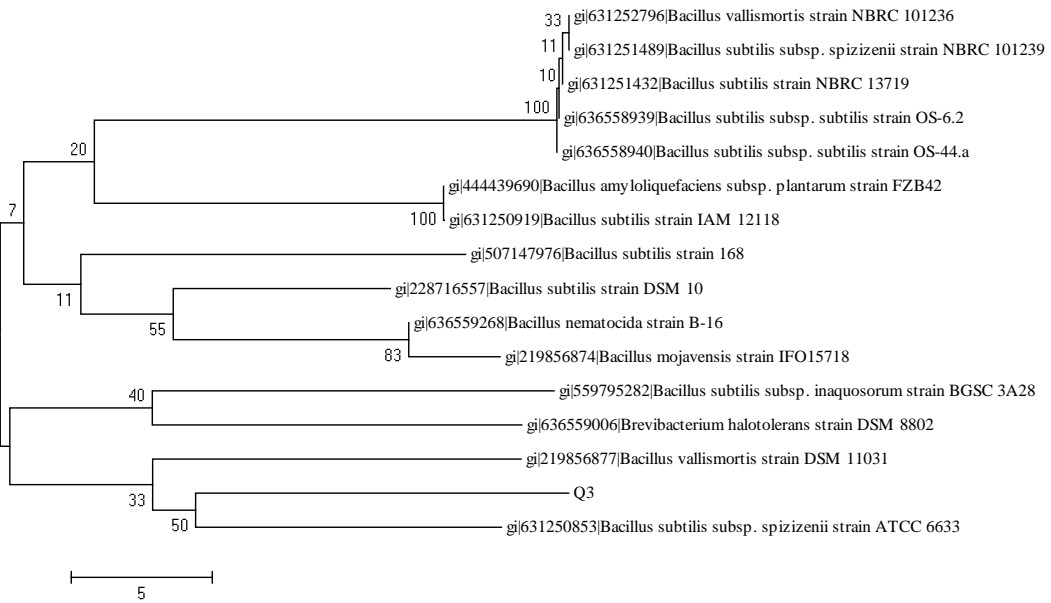

**Figure 9  Neighbor-joining phylogenetic tree of *Bacillus subtilis* strain Q3.** Neighbor-joining phylogenetic tree produced using multiple alignment of 16S rDNA gene sequence of *Bacillus subtilis* strain Q3 with those of other bacterial strains found in the GenBank database.

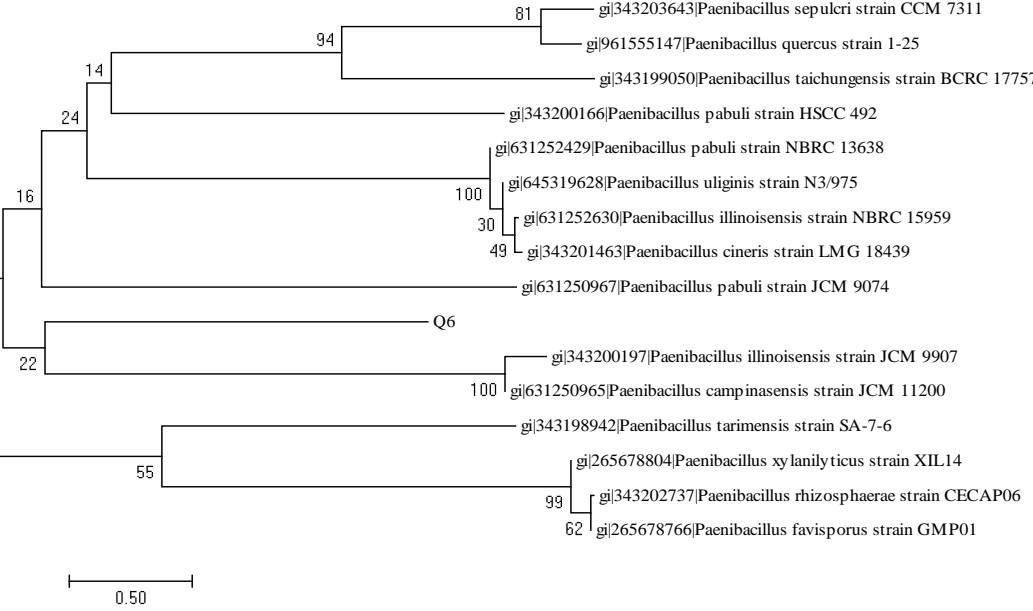

**Figure 10  Neighbor-joining phylogenetic tree of *Paenibacillus* sp. strain Q6.** Neighbor-joining phylogenetic tree produced using multiple alignment of 16S rDNA gene sequence of *Paenibacillus* sp. strain Q6 with those of other bacterial strains found in GenBank database.

cotton crop. As such, these are novel strains which could be further evaluated as potential candidates for improving cotton productivity under alkaline calcareous soil conditions in pot and field experiments to foster growth performances of cotton. There is need to investigate the role of PGPR in fertilizer use efficiency. A better and timelier P uptake by crops may not only reduce input costs but may also save limited resources and decrease eutrophication of water bodies.

## ACKNOWLEDGEMENTS

The authors acknowledge the services of Mikenna Smith for editing the manuscript.

### Funding
Financial support of the research project was provided by the University College of Agriculture and Environmental Sciences and the Islamia University Bahawalpur-Pakistan. The funders had no role in study design, data collection and analysis, decision to publish, or preparation of the manuscript.

### Grant Disclosures
The following grant information was disclosed by the authors:
University College of Agriculture and Environmental Sciences.
Islamia University Bahawalpur-Pakistan.

### Competing Interests
The authors declare there are no competing interests.

### Author Contributions

- Maqshoof Ahmad conceived and designed the experiments, analyzed the data, prepared figures and/or tables, authored or reviewed drafts of the paper, approved the final draft.
- Iqra Ahmad and Muhammad F. Akhtar conceived and designed the experiments, performed the experiments, prepared figures and/or tables, authored or reviewed drafts of the paper, approved the final draft.
- Thomas H. Hilger conceived and designed the experiments, contributed reagents/-materials/analysis tools, authored or reviewed drafts of the paper, approved the final draft.
- Sajid M. Nadeem analyzed the data, authored or reviewed drafts of the paper, approved the final draft.
- Moazzam Jamil conceived and designed the experiments, performed the experiments, analyzed the data, contributed reagents/materials/analysis tools, authored or reviewed drafts of the paper, approved the final draft.
- Azhar Hussain performed the experiments, authored or reviewed drafts of the paper.
- Zahir A. Zahir conceived and designed the experiments, analyzed the data, contributed reagents/materials/analysis tools, authored or reviewed drafts of the paper, approved the final draft.

## DNA Deposition

The following information was supplied regarding the deposition of DNA sequences:
   GeneBank at NCBI database, accession numbers: KX788864, KX788865.

## Data Availability

   The raw data are provided as Supplemental Files.

## Supplemental Information

Supplemental information for this article can be found online at http://dx.doi.org/10.7717/peerj.5122#supplemental-information.

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
