# Peer review of "Preliminary study on phosphate solubilizing Bacillus subtilis strain Q3 and Paenibacillus sp. strain Q6 for improving cotton growth under alkaline conditions"

_PeerJ, doi:10.7717/peerj.5122_

## Round 0.1 · original submission · Major Revisions

I have received two reviews of the manuscript reporting the isolation of P-solubilizing bacteria and the potential of those isolates to promote cotton growth. These reviewers both described the work as “preliminary” but interesting and recommended major revisions. In addition to their comments please consider the following suggestions if you decide to resubmit.

Regards,

Michael

The text requires substantial revision for clarity. It appears that the goal of the project was to test the hypothesis that calcareous soils are a good source of PGPR but this is not made clear. Introduction should be revised to three paragraphs.

1. Phosphorous limits productivity world wide and in Pakistan in particular
2. PGPR can make P available and also have beneficial traits
3. To test the hypothesis…
L 17. Replace “is the problem” with “limits crop production”
L 21. Replace “under laboratory conditions” with “in vitro”
L 25. Here and throughout limit the use of “The” (see also L 131
L 37. That these strains are novel is not convincing. Also, revise to “, which tolerate salinity and solubilize phosphate
L 100. Explain, in the Introduction, why catalase activity was measured and what is the substrate from which ammonium is produced.
L 135. Delete “showed promise and”
L 179. Delete “Twenty.. , Pictures 1 -4) and replace with “All 20 strains we isolated were positive for catalase. One strain (Q19) was positive for urease”
L 194. Delete “The results…that”
L 195. Replace “but” with “with”

·

Basic reporting

Suggest to include "Preliminary study" on the title.
Line 205. I think it will be better to place "%" unit in each of the value.

Experimental design

Line 95. Please include details of the general-purpose agar used. Eg. manufacturer etc.
Line 124-126. Please include the time used to shake the root tips on an orbital shaker.
Line 134-146. What experimental design were used in the in-vitro experiment? How many replicates? Where was the experiment conducted? Growth chamber model?
Line 142. What modifications were done on the Hoagland's solution? Please state.
Line 143. What rock phosphate was used?
Line 160-172. Please consider to eliminate or to redo the phylogenetic tree. A phylogenetic tree is used to show the relationship between the strains and a ref strain of choice, and is therefore shown in only one tree/figure.

Table 2,3,4. It will be better to include the grouping information in the tables. If the treatments did not showed any significant differences, please indicate in the tables.

Validity of the findings

This research has it's merit as a preliminary paper. This paper will be significantly improved if the authors can include a pot study of the Bacillus subtilis strain Q3 and Paenibacillus sp. strain Q6 on cotton, possible a short 1 month study so that a correlation between the parameters can be drawn.
There are improvements that can be made on the data, as mentioned in the experimental design comment section.

Reviewer 2 ·

Basic reporting

I only suggest a reference for introduction (see general comments).

Experimental design

This work is original and very interesting because shown some easy potential strategies to capture bacteria that could increase the growth of cotton. However, in my opinion the information presented is very preliminary yet, especially in experiments in association to seeds and plants. Several aspects of the article are not enough explained, for example there are not figure legends, replica numbers are not showed, methodology should be described more detailed in some cases like in osmoadaptation assays, where authors should mention what they measured in this experiment (optical density?). The problem with optical density is that bacteria under saline stress could express genes in different form and produce minor quantity of secondary metabilites and therefore diminish the value, not nessesary these diminution is linked to bacterial growth.

Validity of the findings

In my opinion the information presented is very preliminary yet, especially in experiments in association to seeds and plants. The number of replicates for plant experiments apparently is very low. Authors in the excel file shown only three replicates for parameters measured, therefore results are not enough for made a conclusion. For bacterial numbers authors apparently assay only two times for colonization or maybe only one for germination and they did not do a corroboration of inoculated strains. This point is essential because today we known that bacteria could be living inside the seeds, therefore is important corroborate that colonization observed is due to inoculated strain and not due to natural population stimulated by inoculated strain. Authors not shown dry weight data which is essential to reinforce conclusions for growth promotion results. Therefore, authors should complete the work before submit again this work for publication.

Additional comments

Dear Michael LaMontagne
Member of the Editorial Board
Of Journal PeerJ

I carefully reviewed the manuscript ID 24744 entitled “Phosphate solubilizing Bacillus subtilis strain Q3 and Paenibacillus sp. strain Q6: High potential candidates for promoting plant growth in cotton under alkaline conditions”. Authors of the present work carried out the isolation of 20 strains from cotton rhizosphere, these strains were screened in some characteristics linked to plant growth promotion like catalase activity, ammonia production and phosphate solubilization, growth under alkaline pH. Seventh strains were selected because they showed all the explored characteristics and were used to explore their ability to colonize and promote the growth of cotton plants. Two strains were the most promising to be used as possible plant growth-promoting bacteria in alkaline fields to increase yield production of cotton. Those strains were partially characterized using molecular tools (16S rDNA sequencing) as Paenibacillus sp. and Bacillus subtillis.
This work is original and very interesting because shown some easy potential strategies to capture bacteria that could increase the growth of cotton. However, in my opinion the information presented is very preliminary yet, especially in experiments in association to seeds and plants. First, Several aspects of the article are not enough explained, for example there are not figure legends, replica numbers are not showed, methodology should be described more detailed in some cases. Second, the number of replicates for plant experiments apparently is very low. Authors in the excel file shown only three replicates for parameters measured, therefore results are not enough for made a conclusion. Third, for bacterial numbers authors apparently assay only two times for colonization or maybe only one for germination and they did not do a corroboration of inoculated strains. This point is essential because today we known that bacteria could be living inside the seeds, therefore is important corroborate that colonization observed is due to inoculated strain and not due to natural population stimulated by inoculated strain. Fourth, authors not shown dry weight data which is essential to reinforce conclusions for growth promotion results. Therefore, authors should complete the work before submit again this work for publication.
Other minor questions and suggestions.
1) In all the paper authors say “16S rRNA sequencing”, but they carried out the amplification of DNA and they did not made a RT-PCR. Therefore this should be substituted per “16S rDNA sequencing”
2) Introduction section. Authors wrote “…(4) production of phytohormones such as indole-3-acetic acid (IAA) (Gupta et al., 2000; Fernando et al., 2006). PGPR are also able to produce plant growth regulators like indole acetic acid, gibberellic acid, cytokinins and ethylene as well as change their concentrations in the rhizosphere (Beneduzi et al., 2008).” Please increase the quality of the text.
3) Introduction. I suggest a reference in the phrase: “When selecting the best-suited PGPR to a given site, a combination of two or more traits is a much more promising approach compared to the use of a single character (Zahir et al., 2003; Ahmad et al., 2008; Baez-Rogelio et al., 2017).”
Baez-Rogelio A., Morales-García Y. E., Quintero-Hernández V., Muñoz-Rojas J. 2017. Next generation of microbial inoculants for agriculture and bioremediation. Microbial Biotechnology 10(1): 19-21.
4) Material and methods, section Isolation of rhizobacteria from cotton rhizosphere. What does mean general-purpose agar plates? Please clarify.
5) What did authors measure in the osmoadaptation assays experiment? Please clarify.
6) Material and methods, section Growth of PGPR at different pH levels. Use international unities, therefore, change “24 hrs” per “24 h”
7) Why the bacterial inoculation time was different for “Root colonization assay” (10 minutes) than that for “Evaluation of phosphate solubilizing PGPR under axenic conditions” (5 minutes). Please clarify.
8) Material and methods, section “Evaluation of phosphate solubilizing PGPR under axenic conditions”. Authors should mention the parameters used to measure the growth of plants, additionally is important corroborate the identity of bacteria quantified in colonization assays.
9) Results, section “Characterization”. In the phrase “Twenty strains were isolated from the rhizosphere of cotton. These were characterized for plant growth promoting traits (Table 1, Pictures 1-4). All of the strains showed positive results for catalase activity by producing gas bubbles;”, I suggest eliminate Pictures 1-4 and “…by producing gas bubbles”, that information is not relevant. Also eliminate the file Picture 1-4.
10) Results, section Identification of selected strains through 16S rRNA sequencing. In my opinion the strain called Bacillus subtilis shod be changed by Bacillus sp. with base to rules of polyphasic taxonomy (16S rDNA is not enough to assignee the bacterial species).
11) Discussion. After the first paragraph authors should explain why they use the catalase assay. By the way, two times the word catalase was bad redacted “catalyse positive”.

---

## Round 0.2 · accepted · Accept

Both reviewers recommended acceptance but I think there are some minor revisions needed that can be incorporated during production. I am heading out of town, so it will take me a few days to send my final edits.

Regards,

Michael

# ·

Basic reporting

no comment

Experimental design

no comment

Validity of the findings

no comment

Additional comments

The author has addressed the previously suggested improvements.

Reviewer 2 ·

Basic reporting

no comment

Experimental design

no comment

Validity of the findings

no comment

Additional comments

Dear Michael LaMontagne
Academic Editor
Peer J
In my opinion all questions were answered satisfactorily and this version of the manuscript was increased in quality and good preliminary results. I have not more comments.